# Tracking Control for an Electro-Hydraulic Rotary Actuator Using Fractional Order Fuzzy PID Controller

**Tri Cuong Do [1], Duc Thien Tran [1,2], Truong Quang Dinh [3] and Kyoung Kwan Ahn [1,*]**

[1] School of Mechanical Engineering, University of Ulsan, 93, Deahak-ro, Nam-gu, Ulsan 44610, Korea; cuongdt298@gmail.com

[2] Department of Automatic Control, Ho Chi Minh City University of Technology and Education, Ho Chi Minh City 700000, Vietnam; thientd@hcmute.edu.vn

[3] Warwick Manufacturing Group (WMG), University of Warwick, Coventry CV4 7AL, UK; t.dinh@warwick.ac.uk

[*] Correspondence: kkahn@ulsan.ac.kr; Tel.: +82-52-259-2282

**Abstract:** This paper presents a strategy for a fractional order fuzzy proportional integral derivative controller (FOFPID) controller for trajectory-tracking control of an electro-hydraulic rotary actuator (EHRA) under variant working requirements. The proposed controller is based on a combination of a fractional order PID (FOPID) controller and a fuzzy logic system. In detail, the FOPID with extension from the integer order to non-integer order of integral and derivative functions helps to improve tracking, robustness and stability of the control system. A fuzzy logic control system is designed to adjust the FOPID parameters according to time-variant working conditions. To evaluate the proposed controller, co-simulations (using AMESim and MATLAB) and real-time experiments have been conducted. The results show the effectiveness of the proposed approach compared to other typical controllers.

**Keywords:** hydraulic system; electro-hydraulic actuator; fractional order PID; fuzzy logic system

## 1. Introduction

Considering the improvement of industry, robotics and smart systems are becoming increasingly popular and widely used. Among them, hydraulic systems are among the preferred options in modern industries due to their advantages such as durability, controllability, accuracy, reliability, price [1–5]. An electro-hydraulic actuator (EHA) system is known as a typical hydraulic system and is employed to overcome the problems of the conventional hydraulic system where actuator depends on the state of the main control valve caused by inefficiency and loss of energy during the operation process [6,7]. In detail, the EHA contains a hydraulic power pack (a bi-directional pump, an electric motor and a reservoir), supplement valves, and an actuator. The system does not include a control valve, which reduces pressure losses and heat generation in the valve. However, the main weaknesses of the EHA system are its complex dynamics, high non-linearity and high uncertainty due to the instability of some hydraulic parameters that make it difficult to control.

The conventional proportional integral derivative (CPID) control algorithm (integral-derivative ratio) is recognized as the most common method used in industrial process control because of its simple structure, feasibility and ease of implementation. Hence, some authors applied a conventional PID (CPID) controller on the EHA system. Navatha et al. [8] used a conventional PID (CPID) to analyze the dynamic, position tracking and control of the EHA system. PID tuning has been done

using the Ziegler Nichols method. To improve the performance of the EHA system, Ha et al. [9] proposed an adaptive PID based on sliding mode to control the non-linearity and uncertainty factors. Truong et al. [10] suggested a grey prediction model combined with a fuzzy PID controller. In detail, fuzzy controllers and a tuning algorithm adjusted the grey step size. The grey prediction compensator can reduce settling time and overshoot problems. Nevertheless, the conventional PID (CPID) controller has some disadvantages such as error calculation: the step reference signal is often used and the CPID demands a large control signal to perform it, amplification of noise, oversimplification, and complexity due to integral control [11–13]. Therefore, the CPID controller becomes inefficient with the highly non-linear system possessing unclear behavior, particularly in an EHA system. With the aim of attaining more favorable dynamic performance and the stability of the controlled systems, Podlubny has proposed a new controller called a fractional order PID (FOPID) controller [14]. The control performance for this controller is built on the theory of fractional order calculation including the $\lambda$ (non-integer order of integrator) and $\mu$ (non-integer order of differentiator) parameters. However, with the expansion of the calculation area, the FOPID controller has a total of five parameters ($K_p$, $K_i$, $K_d$, $\lambda$, $\mu$) that need to be determined and this is a challenge for the designer. In order to solve this problem, several intelligent methods such as neural network, fuzzy logic, and optimization methods, were merged with FOPID controller and then these methods adjust the parameters of FOPID controller depending on the working conditions [15–17]. Although these approaches could improve the control performance, the control complexity is considered the key enabler. Among these techniques, fuzzy-based FOPID control offered the simplest solutions whilst ensuring the effectiveness of the FOPID controller [18–21]. The fuzzy logic controller (FLC) emulates human thinking and can be tuned clearly to acquire the ideal performance of the control system online without the accurate mathematical model of the controlled objective. However, the controllers presented in the previous studies only designed a rule for the fuzzy logic system with two inputs which were errors and the derivative of errors and only one output. In fact, the FOPID controller has five different parameters ($K_p$, $K_i$, $K_d$, $\lambda$, $\mu$) and each parameter has a different effect on the controller's performance. Therefore, each parameter needs to have its own design rules to find the best parameters during operation.

Based on the previous investigation, this paper presents an efficient controller via a combination between the FOPID controller and a fuzzy logic system for position control of a loading system using an electro-hydraulic rotary actuator (EHRA) which is a type of EHA system. The FOPID controller is used to enhance the tracking performance of the EHRA system. Besides, the FLC with 2 inputs (errors and derivative of errors) and 3 outputs along with 3 separate rules is designed to adjust the controller parameters ($K_p$, $K_i$, $K_d$). The $\lambda$ (non-integer order of integrator) and $\mu$ (non-integer order of differentiator) parameters are determined by the trial-error method and kept constant during the operations. Several experiments and simulations of EHRA with PID, fuzzy PID (FPID), FOPID and fractional order fuzzy PID (FOFPID) are investigated in variant functioning requirements (adjusted references, operating frequencies, and variable external weights). The results illustrate that the proposed FOFPID controller accomplishes better performance with more precision under numerous operating circumstances and strong applicability in present hydraulic systems.

The rest of this paper is arranged as follows: Section 2 studies the detailed description and the dynamical mechanism equations of the loading system using EHRA. The controller design method is introduced in Section 3. The simulation and experiment results are contributed in Section 4. In the final section, some conclusions are summarized.

## 2. System Configuration

The loading system using EHRA in this paper includes a gear pump, supplement valves, a hydraulic rotary actuator, pulley, cable and load as presented in Figure 1. The bi-directional rotational pump is used and driven by the direct current (DC) servo motor so that the hydraulic oil line from the pump can be supplied directly to the actuator without a control valve in both directions. The controlled motor speed which meets the system requirements (flow rate and pressure) can reduce the

power consumption, loss energy and heat generation. In addition, the supplement valves are well equipped as a safety function during the lifting and lowering processes.

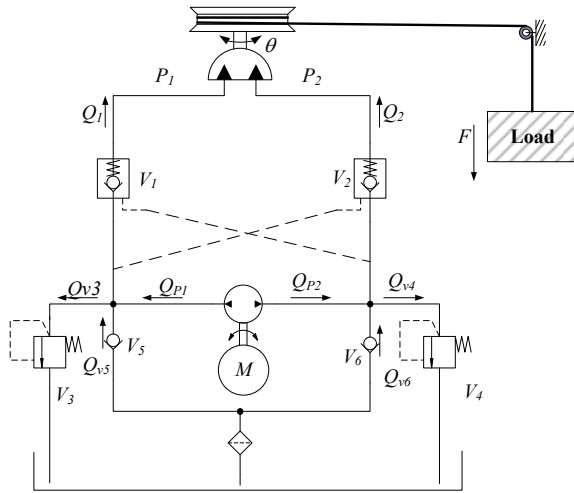

**Figure 1.** Configuration of a loading system using electro-hydraulic rotary actuator (EHRA).

Based on the system setup in Figure 1, by using the second Newton's law and principles of hydraulics system, the dynamics of the rotary actuator (RA) can be described by the following state space [22]:

$$J\ddot{\theta} = (P_1 - P_2)D_R - T \tag{1}$$

where $\ddot{\theta}$ is the loading system's rotor angular acceleration, $J$ is the inertia moment of load shaft, $D_R$ is the displacement of the rotary actuator, $P_i (i = 1,2)$ is the pressure in both side chambers of the RA, $T$ is the torque at the output shaft RA.

Supplied flow rates into both side chambers are calculated as:

$$\begin{cases} Q_1 = Q_{P1} + Q_{v5} - Q_{v3} \\ Q_2 = Q_{P2} + Q_{v6} - Q_{v4} \end{cases} \tag{2}$$

where $Q_{P1} = -Q_{P2} = Q_{pump}$ is the supplied flow rate from the main pump, and the terms $Q_{vi} (i = 3,..,6)$ are flow rates through valves $v_i (i = 3,..,6)$, respectively.

$$Q_{pump} = D\omega - k_{leakage}(P_1 - P_2) \tag{3}$$

where $D$ is the displacement of the pump, $k_{leakage}$ is the leakage coefficient and $\omega$ is the velocity of the DC motor.

During normal working conditions, the pressure values in the two chambers of rotary: $P_1, P_2$ should be maintained lower than the setting pressure value $P_{set}$ of the relief valves: $V_3$, and $V_4$ and these relief valves are closed. Then, Equation (2) can be modified as:

$$\begin{cases} Q_1 = Q_{P1} + Q_{v5} \\ Q_2 = Q_{P2} + Q_{v6} \end{cases} \tag{4}$$

We assume that the external leakage has not happened and the dynamics of oil flow can be computed:

$$\begin{cases} \dot{P_1} = \dfrac{\beta}{V_{01} + A\theta}\left(Q_1 - A\dot{\theta} - C_t\left(P_1 - P_2\right)\right) \\ \dot{P_2} = \dfrac{\beta}{V_{02} - A\theta}\left(Q_2 + A\dot{\theta} + C_t\left(P_1 - P_2\right)\right) \end{cases} \tag{5}$$

where $V_i$ $(i = 1, 2)$ is total volumes of two chambers, $\dot{\theta}$ and $C_t$ is the shaft speed and the coefficient of the internal leakage of the RA.

The system states can be defined as:

$$x = (x_1, x_2, x_3, x_4,)^T = (\theta, \dot{\theta}, P_1, P_2,)^T \tag{6}$$

Gathering Equations (1)–(5), the state space of the EHRA system can be presented as follows:

$$\begin{cases} \dot{x}_1 = x_2 \\ \dot{x}_2 = \dfrac{D_R}{J}(x_3 - x_4) - \dfrac{T}{J} \\ \dot{x}_3 = \dfrac{\beta_e}{V_{01} + Ax_1}\left(D\omega - \left(k_{leakage} + C_t\right)(x_3 - x_4) + Q_{v1} - Ax_2\right) \\ \dot{x}_4 = \dfrac{\beta_e}{V_{02} - Ax_1}\left(-D\omega + \left(k_{leakage} + C_t\right)(x_3 - x_4) - Q_{v2} + Ax_2\right) \end{cases} \tag{7}$$

To make easier the state space of the system (7), we describe:

$$x_{34} = \frac{x_3 - x_4}{J}D_R ; d_1 = -\frac{T}{J}; k_{leak} = k_{leakage} + C_t; \beta = \frac{\beta_e}{J}$$

$$g(x_1) = \left(\frac{\beta}{V_{01} + Ax_1} + \frac{\beta}{V_{02} - Ax_1}\right)D_R D$$

$$f(x_1, x_2) = -\beta D_R Ax_2\left(\frac{1}{V_{01} + Ax_1} + \frac{1}{V_{02} - Ax_1}\right)$$

The rotation speed of the bi-directional pump driven by a DC motor adjusts the system states. A bounded desired trajectory is given: $x_{1d}$. Therefore, the target of this paper is to regulate the input velocity demand for a DC motor $\omega$ to manipulate the output position $x_1$ tracks closely as possible to the desired reference. Then the state space (7) can be characterized in harsh feedback form:

$$\begin{aligned} \dot{x}_1 &= x_2 \\ \dot{x}_2 &= x_{34} + d_1(t) \\ \dot{x}_{34} &= g(x_1)u + f(x_1, x_2) + d_2(t) \end{aligned} \tag{8}$$

The matched and mismatched disturbances $di(t)$ $(i = 1, 2)$, their first derivatives and their second derivatives are bounded.

$$\left|d_i(t)\right| \le \kappa_i, \left|\dot{d}_i(t)\right| \le \alpha_i, \left|\ddot{d}_i(t)\right| \le \beta_i \tag{9}$$

where $\kappa_i$, $\alpha_i$ and $\beta_i$ are positive constants.

## 3. Controller Design for Electro-Hydraulic Rotary Actuator (EHRA) System

In this paper, the main task is to guarantee the angle position of RA follows the required trajectory output as much as possible. Therefore, to achieve this obligation, a fractional order fuzzy PID (FOFPID) controller is performed with the overall structure shown in Figure 2.

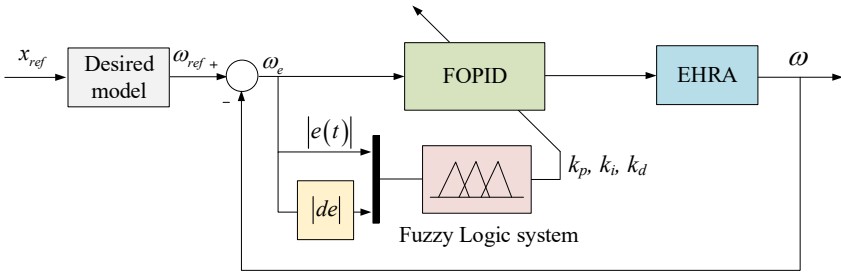

**Figure 2.** Block diagram of the proposed fractional order fuzzy PID (FOFPID) controller.

### 3.1. Fractional Order Calculation

Fractional calculus is used three centuries ago, but it is not very popular or widely applied in research fields. In recent years, a lot of researchers have achieved remarkable achievement in many different areas such as control system, speech signal processing or modelling using fractional calculus [23]. Fractional calculus is a generalization of integration and differentiation to non-integer order operator, where $a$ and $t$ denote the limits of the operation and $\alpha$ denotes the fractional order such that:

$$_aD_t^\alpha = \begin{cases} \dfrac{d^\alpha}{dt^\alpha} & \Re(\alpha) > 0 \\ 1 & \Re(\alpha) = 0 \\ \displaystyle\int_a^t (dt)^{-\alpha} & \Re(\alpha) < 0 \end{cases} \tag{10}$$

where generally it is assumed, that $\alpha \in \mathbb{R}$. There exist many definitions of the fractional calculus. There are three main definitions namely Grunwald–Letnikov, Riemann–Liouville, and the Caputo definition. Among them, Reimann–Liouville's differ integral (RL) definition is widely used. It is defined as the following:

$$_aD_t^\alpha = D^n J^{n-\alpha} f(\mathrm{t}) = \frac{1}{\Gamma(\mathrm{n}-\alpha)} \left(\frac{d}{dt}\right)^n \int_a^t \frac{f(\tau)}{(\mathrm{t}-\tau)^{\alpha-n+1}} d\tau \tag{11}$$

where $n$ is the integer value which satisfies the condition $n-1 < \alpha < n$, $\alpha$ is a real number, $J$ is the integral operator. The Gamma function used in the above Equation (11) can be defined by the following:

$$\Gamma(\mathrm{x}) = \int_0^\infty t^{x-1} e^{-t} dt, \Re(\mathrm{x}) > 0 \tag{12}$$

**Remark 1.** *In this paper, the Riemann Liouville definition is used for fractional integral and derivative calculation. The fractional-order modeling and control (FOMCON) toolbox which is developed by Aleksei Tepljakov [24] is employed in MATLAB/Simulink platform to simulate the FOPID controller.*

### 3.2. Fractional Order PID Controller

The calculation equation of the fractional order PID controller applied for the loading system can be presented in the time domain by:

$$U(n) = K_P(n)e(t) + K_I(n)\frac{d^{-\lambda}e(t)}{dt^{-\lambda}} + K_D(n)\frac{d^\mu e(t)}{dt^\mu} \tag{13}$$

where the control signal, $U(n)$, is the velocity command of the DC motor. $K_p$, $K_i$, and $K_d$ are the proportional, the integral, and the derivative coefficients, respectively. The error between the actual position $x_1$ getting from the sensors and the desired trajectory $x_{ref}$ is defined as follows:

$$\mathrm{e}(t) = x_{ref} - x_1(t) \tag{14}$$

All the CPID controllers are particular cases of the fractional controller, where $\lambda$ and $\mu$ are equal to one. In the FOPID controller, the order of the elements I and D is not only equal to one but also can change over a wider range from zero to two refer to Figure 3. Besides setting the proportional, derivative and integral gains $K_p$, $K_i$, $K_d$, two additional parameters (the order of fractional integration $\lambda$ and fractional derivative $\mu$) also have to be specified. By expanding the calculation region of derivation and integration based on the fractional order theory, the scale of the controller parameters setting becomes larger and the controllers become more flexible and stable to the controlled objectives, and the system performance can also be enhanced at the same time.

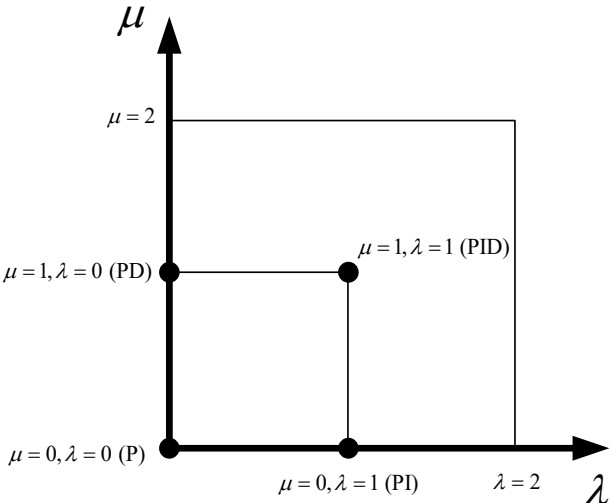

**Figure 3.** The converge of fractional order PID (FOPID) controller.

### 3.3. Fractional Order Fuzzy PID Controller Design

Generally, the fuzzy logic system is designed based on system characteristics and control purpose. From Equation (13), three parameters $K_p$, $K_i$, $K_d$ of FOPID controller are regulated by using the fuzzy logic system. Subsequently, the overall proposed fractional order fuzzy PID controller is made up of a combination between three separate fuzzy P, I, and D functions and the FOPID controller. The applied fuzzy scheme details for the EHRA system is presented in Figure 4. The fuzzy logic system contains two input signals: absolute error and absolute derivative of the error. These input signals are scaled in the range 0 to 1, which are derived from the difference between the actual position and the desired trajectory of the system. Inside the signal blocks, four membership functions, namely 'S', 'M', 'B' and 'VB', divide the input signal into four evenly spaced intervals corresponding to the four values 'small', 'medium', 'big' and 'very big' of the error as presented in Figure 5. Similar to the input signal, three output signals that correspond to the $k_p$, $k_i$ and $k_d$ values are also selected from 0 to 1 and divided into equal intervals into the output fuzzy blocks. Based on the above embedded membership function, the three fuzzy rules are established to adjust the output values following the input values and are listed in Tables 1–3. The fuzzy rules are composed as follows

**Rule i.** *If the input values $\left|e(t)\right|$ is $A_i$ and $\left|de(t)\right|$ is $B_i$ then the output values $k_p$ coefficient is $C_i$, $k_i$ coefficient is $D_i$ and $k_d$ coefficient is $E_i$ ($i = 1, 2, \ldots, n$).*

where $n$ is the number of fuzzy rules; $A_i$, $B_i$, $C_i$, $D_i$ and $E_i$ are the $i$th fuzzy sets of the input and output variables used in the fuzzy rules. $A_i$, $B_i$, $C_i$, $D_i$ and $E_i$ are also the variable values $k_p$, $k_i$ and $k_d$, respectively. The output values are obtained by the collection operation of set fuzzy inputs and the created fuzzy rules, where the MAX–MIN aggregation method and 'centroid' defuzzification method are employed. Finally, these output values are replaced in the following Equation (15) to estimate the factors $K_p$, $K_i$ and $K_d$:

$$\begin{cases} k_p = \dfrac{K_p - K_{p\min}}{K_{p\max} - K_{p\min}} \\[2mm] k_i = \dfrac{K_i - K_{i\min}}{K_{i\max} - K_{i\min}} \\[2mm] k_d = \dfrac{K_d - K_{d\min}}{K_{d\max} - K_{d\min}} \end{cases} \tag{15}$$

The ranges of $K_p$, $K_i$ and $K_d$ are defined as $[K_{pmin}, K_{pmax}]$, $[K_{imin}, K_{imax}]$, and $[K_{dmin}, K_{dmax}]$, respectively.

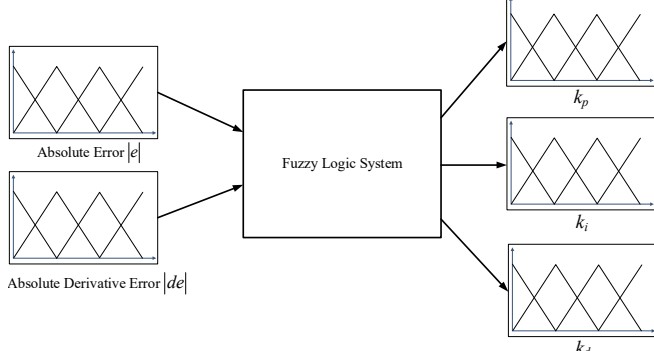

**Figure 4.** Fuzzy design for tuning parameters ($K_p$, $K_i$, $K_d$).

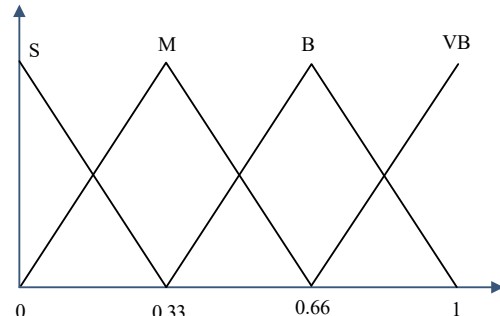

**Figure 5.** Setup membership functions of the inputs and output fuzzy logic system.

**Table 1.** Fuzzy rule of $K_p$.

| $\|de(t)\|$ | $\|e(t)\|$ | | | |
|:---:|:---:|:---:|:---:|:---:|
| | **S** | **M** | **B** | **VB** |
| **S** | M | M | B | VB |
| **M** | S | M | B | VB |
| **B** | S | S | B | B |
| **VB** | S | S | M | B |

**Table 2.** Fuzzy rule of $K_i$.

| $\|de(t)\|$ | $\|e(t)\|$ | | | |
|:---:|:---:|:---:|:---:|:---:|
| | **S** | **M** | **B** | **VB** |
| **S** | VB | VB | S | S |
| **M** | VB | VB | S | S |
| **B** | VB | VB | M | S |
| **VB** | VB | B | M | S |

**Table 3.** Fuzzy rule of $K_d$.

| $\|de(t)\|$ | $\|e(t)\|$ | | | |
|:---:|:---:|:---:|:---:|:---:|
| | **S** | **M** | **B** | **VB** |
| **S** | B | M | S | S |
| **M** | B | B | S | S |
| **B** | VB | B | M | S |
| **VB** | VB | B | M | S |

## 4. Simulation Results

Based on the above analysis, a co-simulation between AMESim 15.2 and MATLAB 2017a was built to prove the effectiveness of the proposed controller as shown in Figure 6. The co-simulation structure contained two parts: the dynamic system was the first part and the controllers were the second part. In detail, the loading system using the EHRA structure was simulated in AMESim 15.2 software in which the models of hydraulic devices were simulated in blue blocks while the mechanical parts were illustrated in blue blocks and control signals are indicated by red lines as presented in Figure 7 Besides, the proposed controller was programmed in the MATLAB/Simulink and imported to the EHRA model through the S function. The system parameters were set according to the real test bench as listed in Table 4.

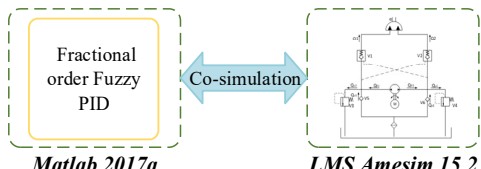

**Figure 6.** Communication between the system dynamic and the proposed controller.

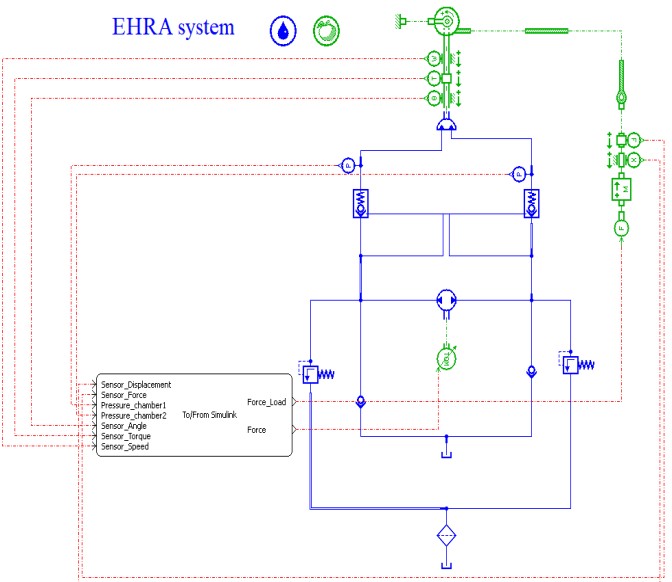

**Figure 7.** The dynamic EHRA system in AMESim software.

Some simulations were conducted in different working conditions for verifying the control performances of the proposed controller with other three controllers, CPID, FPID, and FOPID. First, reference input was the step signal with the amplitude of 15 degrees, and the payload was 150 N. Second, reference input was the multi-step signal with maximum amplitude of 30 degrees, and the payload was increased to 500 N. Third, reference input was the sine signal of 0.05 Hz, and the payload was 50 N. The coefficients of the controllers were selected as follows: CPID: $K_p = -1000$, $K_i = -1$, $K_d = -600$; FPID: $K_{pmin} = -700$, $K_{pmax} = -200$, $K_{imin} = -3$, $K_{imax} = -1$, $K_{dmin} = -40$, $K_{dmax} = -10$; FOPID: $K_p = -1000$, $K_i = -1$, $K_d = -600$, $\lambda = 0.1$, $\mu = 0.7$; Proposed control: $K_{pmin} = -700$, $K_{pmax} = -200$, $K_{imin} = -5$, $K_{imax} = -1$, $K_{dmin} = -600$, $K_{dmax} = -200$, $\lambda = 0.1$, $\mu = 0.7$.

**Remark 2.** *To illustrate the effectiveness of the proposed control, the coefficients of the controllers were selected in the first simulation condition, then they were maintained in other simulation conditions.*

**Table 4.** Specification parameter of the EHRA system.

| Components | Parameters | Value | Unit |
|---|---|---|---|
| Hydraulic Pump | Displacement | 0.97 | cc/rev |
| | Rated rotation speed | | rpm |
| Relief valve | Pressure | 120 | bar |
| Hydraulic Rotary Actuator | Displacement | 27.54 | cc/rev |
| | Rotation angle | 100 | deg. |
| | Torque output | 120 | Nm |
| Hydraulic oil | Effective bulk modulus | $1.5 \times 10^9$ | Pa |
| | Density | 0.87 | kg/dm³ |
| | Viscous Friction Coefficient | 30 | N/(m/s) |
| Encoder | Model | E40H8-5000-3-N-24 | |

In the first simulation, Figure 8a represents the tracking performance of all controllers with the step reference signal. By using the fuzzy logic control system, the FPID controller was able to improve the performance of the conventional controller. When applying the FOPID and FOFPID controller to the system, the accuracy of the EHRA system was significantly increased. The results in Figure 8b validates that the FOFPID controller can reduce the rising time, the settling time and overshoot. The steady-state error of the actual position is within 0.1% of the desired trajectory.

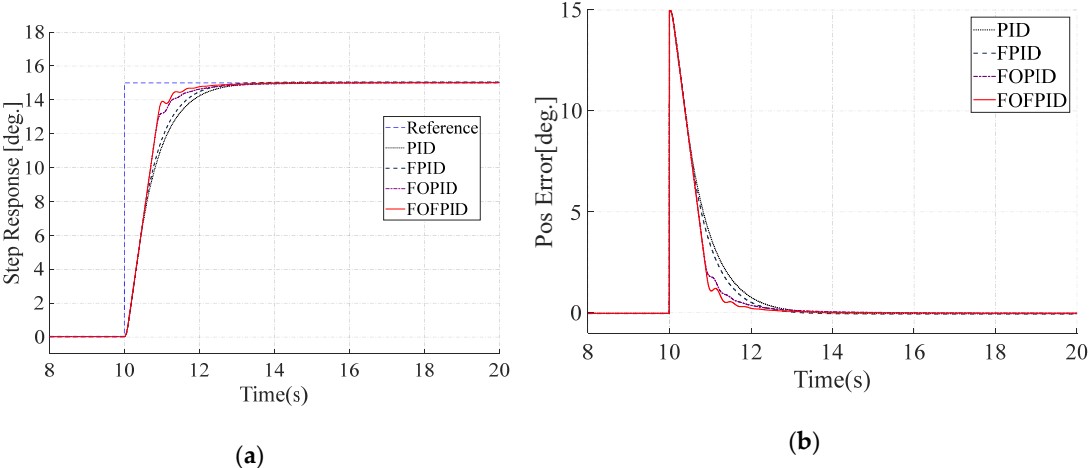

(**a**)                    (**b**)

**Figure 8.** Comparison for step trajectory (amplitude 15 degrees and the payload is 150 N). (**a**) Output performance; (**b**) error effort.

In the second simulations, Figure 9 shows the multi-step responses and error effort of CPID, FPID, FOPID and the proposed controller with the biggest step is 30 degrees and the payload is higher than the previous simulation. It can be seen that the difference in tracking performance of the controllers at the large step signal is not much. However, at the small step values, the proposed FOFPID controller achieves the best implementation with the fastest rising time when compared with the other controllers. This demonstrated that the proposed controller could quickly calculate and choose the parameters, then, given the suitable control signal to the system to meet the reference signal. Besides, the weight acting on the system is also changed but the state error is kept within 0.1 degree.

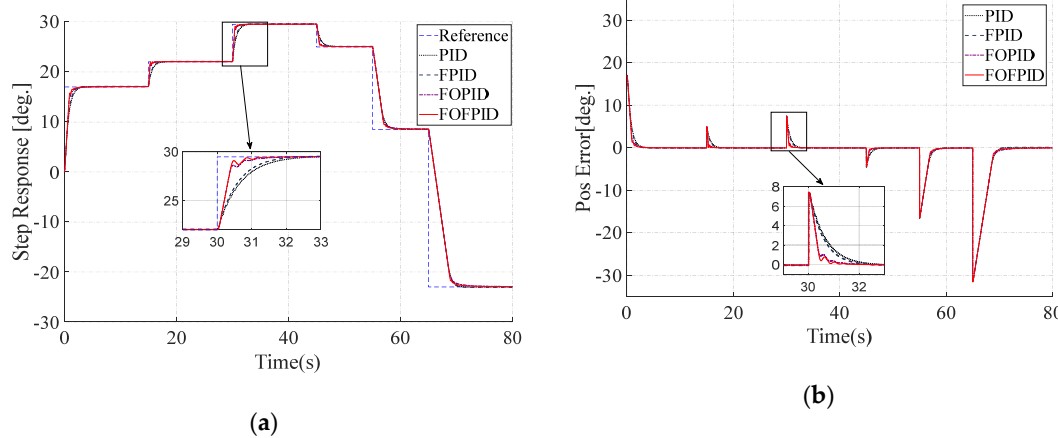

**Figure 9.** Comparison for multi-step trajectory (maximum amplitude 30 degrees and the payload is 500 N). (**a**) Output performance; (**b**) error effort.

In the third simulation, with the purpose of testing the controller under different trajectories, the reference is changed to the sinusoidal signal. The frequency of the sinusoidal reference signal was 0.05 Hz and the payload was 50 N. Figure 10a illustrated the tracking performance of the CPID, FPID, FOPID and FOFPID controllers. The proposed control still achieved the best performance. The result in Figure 10b indicated that the proposed FOFPID controller has the smallest error in the controllers. The error of FOFPID was in the range of [−0.04, 0.04] degree while CPID was from −0.4 to 0.4 degree, FPID is from −0.2 to 0.2 and FOPID is from −0.06 to 0.06 degree.

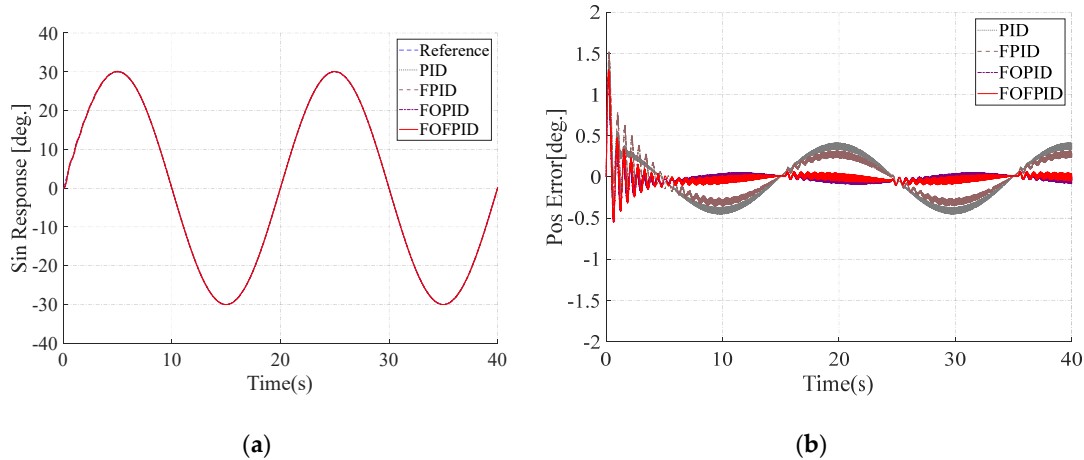

**Figure 10.** Comparison for 0.05 Hz sinusoidal trajectory (the payload is 50 N). (**a**) Output performance; (**b**) error effort.

The conducted simulations demonstrated the effectiveness of the proposed FOFPID controller. However, the simulation may not fully represent actual system parameters such as friction, temperature, the viscosity of hydraulic oil. Therefore, the proposed controller should be tested for a real system to prove its applicability.

## 5. Experimental Results

After successfully testing the effectiveness of the proposed controller on the simulation system, a real EHRA test bench was built. The structure of the experimental loading system using EHRA was presented in Figure 11. The power pack system is chosen from the Bosch Rexroth company, which

contained a bi-directional gear pump, a 24 V-20 A DC motor, a small tank, and some safety valves. This power pack converted the electricity to hydraulic power and supplied to the whole system. In this configuration, the angular rotation of the rotary actuator is controlled by the speed of the DC motor whose speed is adjusted directly from the computer through the PCI card and the motor driver. The rotary actuator with a limited angle rotation (100 degrees) is manufactured by KNR Company (Yongin-si, Gyeonggi-do, Korea). One rotary encoder and two pressure transducers are set up to the system to measure the angle rotation and the pressure in two chambers of the rotary actuator, respectively. The load simulator part is designed as an external gravitational force acting on the loading system which can be easily adjusted by changing the attached mass. This is a simple way to change the working conditions of the system. The proposed controller is programed on the computer within real-time Window Target Toolbox of MATLAB 2013b under a sampling time of 0.01 s. The detailed specifications of the system components are summarized in Table 4, and the real apparatus is shown as Figure 12. By using trial-error method, the control parameters can be selected as follow: CPID: $K_p = -580$, $K_i = -15$, $K_d = -22$; FPID: $K_{pmin} = -600$, $K_{pmax} = -200$, $K_{imin} = -25$, $K_{imax} = -10$, $K_{dmin} = -35$, $K_{dmax} = -10$; FOPID: $K_p = -640$, $K_i = -15$, $K_d = -25$, $\lambda = 0.3$, $\mu = 0.6$; Proposed control: $K_{pmin} = -700$, $K_{pmax} = -200$, $K_{imin} = -25$, $K_{imax} = -10$, $K_{dmin} = -40$, $K_{dmax} = -10$, $\lambda = 0.3$, $\mu = 0.6$.

**Remark 3.** *In practice, because of the mechanical characteristic of the EHRA system, it cannot work stably in high frequency. Therefore, the 9 Hz cutoff frequency is chosen for some testing cases.*

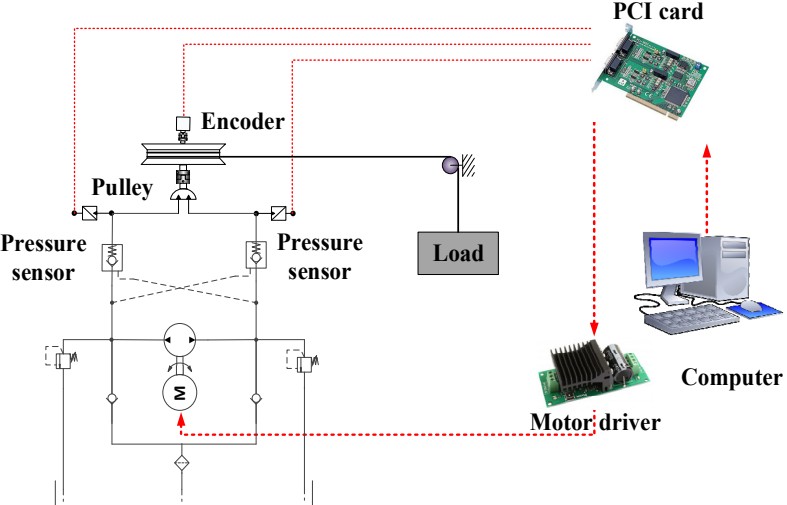

**Figure 11.** Structure of the experimental loading system using EHRA.

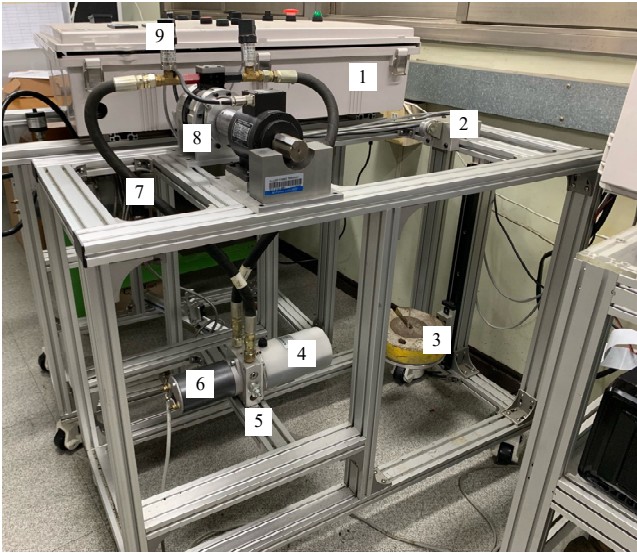

**Figure 12.** Experimental apparatus with 1—Control box, 2—Pulley and cable, 3—Attached weight, 4—Oil tank, 5—Pump and valves, 6—Electric DC motor, 7—Pipeline, 8—Rotary actuator, 9—Pressure sensor.

In this case, the pulse reference signal is used with amplitude [−5, 5], period 10 s, the load 5 kg and cutoff frequency 9 Hz. Figure 13 depicts the comparisons of position response using CPID, FPID, FOPID and proposed FOFPID controllers. From the results, we can see that the FOFPID controller provided the best response. The overshoot is a little bit bigger than the FPID and PID controllers. But the rising time and settling time are shorter. The steady-state error is kept in the range of [−1, 1]. The control signal in comparison showed that the chattering phenomenon caused by derivative element is reduced significantly according to the online adjustment the gains of the proposed controller as can be seen in Figure 14.

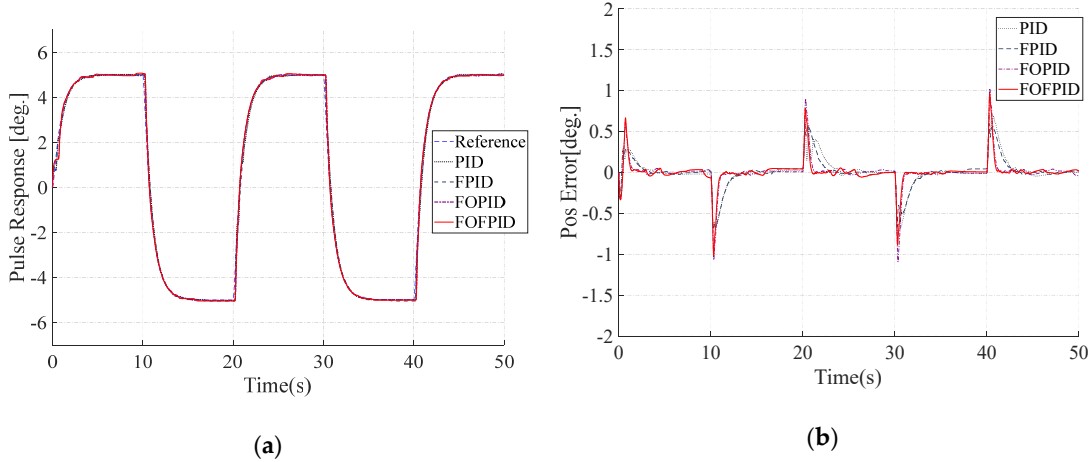

(**a**)

(**b**)

**Figure 13.** Comparison for pulse trajectory (cutoff frequency 9 Hz and the payload is 50 N). (**a**) Output performance; (**b**) error effort.

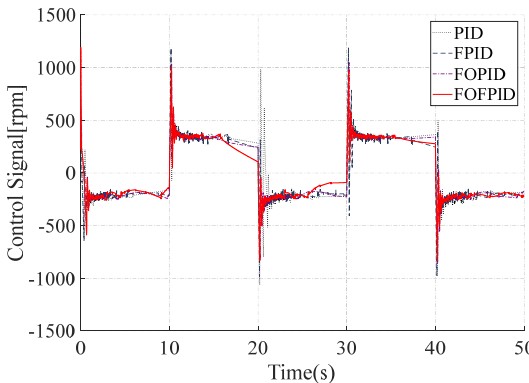

**Figure 14.** Control signal of conventional PID (CPID), fuzzy PID (FPID), FOPID and proposed FOFPID.

In order to investigate more challenging working conditions with the pulse reference signal, the cutoff frequency was changed to 2 Hz. The performances of the controllers are shown in Figure 15a. With the PID and FPID controllers, they could not maintain the accuracy of the system under the dynamic separation of weight at high cutoff frequencies from 10 s. These phenomena did not occur in the simulation results due to the fact that the real test bench has many parameters that could not be determined, and they affected the control quality of the controllers. Meanwhile, the proposed controller could compensate for these problems of the previous controllers and still achieved an ideal result with the fastest rising time and settling time. The steady-state error was also guaranteed and kept in the range from [−1, 1] degrees as could be seen in Figure 15b. Based on these pulse reference signal experiments, the effectiveness of the proposed controller over the conventional PID, fuzzy PID and fractional order controllers were strongly confirmed.

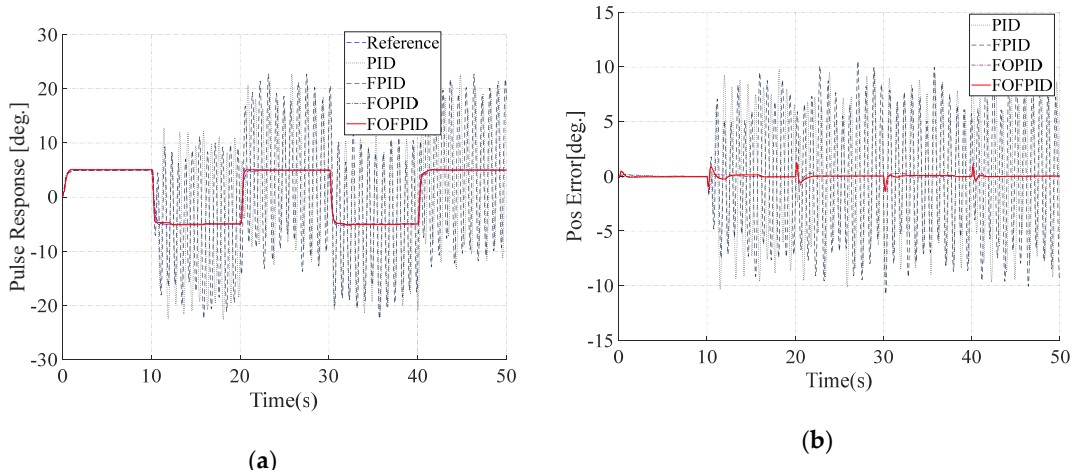

(**a**)

(**b**)

**Figure 15.** Comparison for pulse trajectory (cutoff frequency 2 Hz and the payload is 50 N). (**a**) Output performance; (**b**) error effort.

To further investigate the transient response, the steady-state behavior with non-periodic signals, the multistep reference with a maximum amplitude of 35 degrees was used while the load was set to 5 kg. With the same controllers, the system response and tracking error were presented in Figure 16a,b respectively. The proposed FOFPID provided a fast response with high accuracy of the steady-state control error (within 0.3 degree). The values of $K_p$, $K_i$, $K_d$ were adjusted by fuzzy rule can be seen in Figure 17.

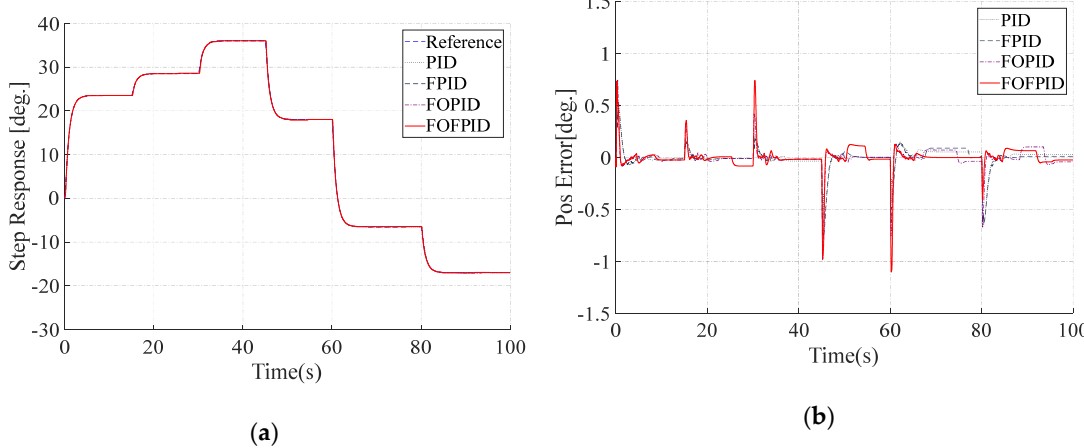

(**a**)                                                                                   (**b**)

**Figure 16.** Comparison for multi-step trajectory (cutoff frequency 9 Hz and the payload is 50 N). (**a**) Output performance; (**b**) error effort.

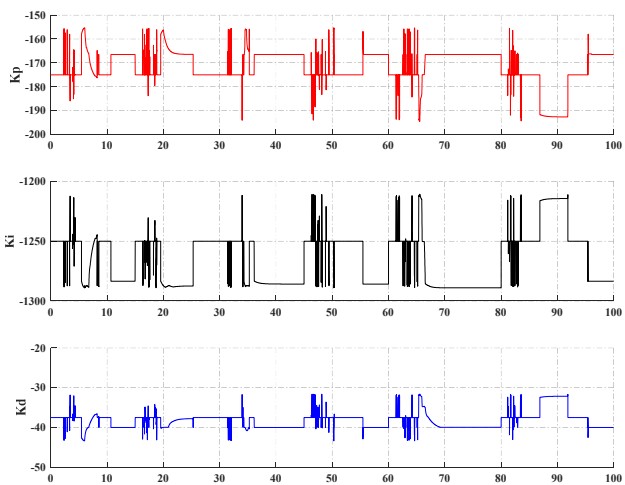

**Figure 17.** Tuning parameters of $K_p$, $K_i$, and $K_d$.

In the final case, the experiment was carried out to deal with big external force condition, the load was changed to 15 kg. The performances of the controllers showed in Figure 18a presented that conventional controllers could not control the system when the rotary reverted its direction to lift the load. Meanwhile, the proposed FOFPID controller could compensate for the impact of external forces on the system and maintained stable performance. The transient error was maintained in the range of [−1, 1] and steady-state error was in the range of [−0.3, 0.3] degree as can be seen in Figure 18b. It was obvious that a good tracking trajectory was investigated when using fractional order and intelligent technique to design the stable position fractional order fuzzy PID controller.

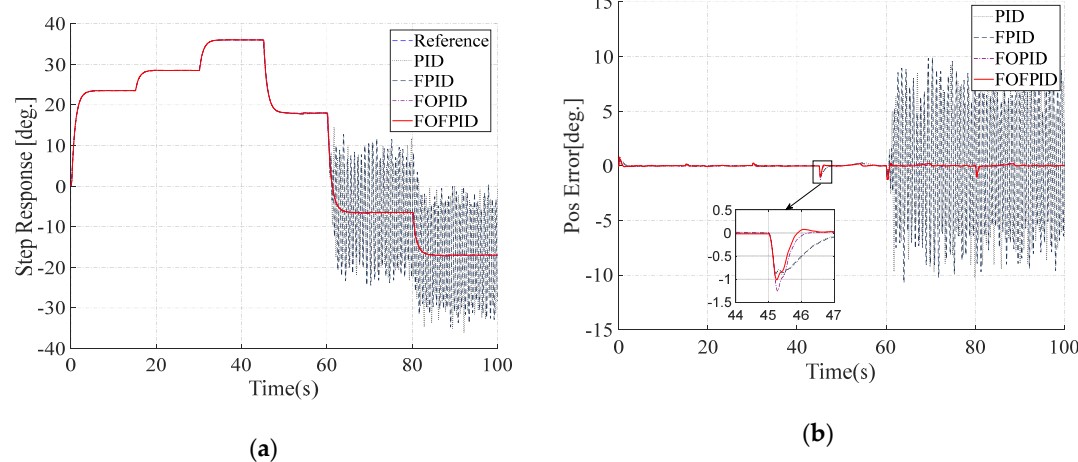

**Figure 18.** Comparison for multi-step trajectory (cutoff frequency 9 Hz and the payload is 150 N). (**a**) Output performance; (**b**) error effort.

## 6. Conclusions

This paper presented an advanced intelligent controller based on the fractional order PID controller combined with the fuzzy logic system for trajectory tracking of the loading system using an electro-hydraulic rotary actuator. The dynamic model of the loading system using EHRA and the controllers was built by using AMESim/MATLAB co-simulation. In addition, a real test bench was fabricated to carry out some tests with respect to different working conditions in order to evaluate the stability performance of the proposed controller. The comparison results indicated that the proposed controller achieved a good tracking trajectory. It could compensate for the changed external force when the rotary actuator changed direction and provided suitable control signals at high cut-off frequencies. Consequently, the proposed FOFPID controller has strong control proficiency not only for applications of the EHRA system but also for other complex requirement systems. In future work, some adaptive laws will be considered to improve the performance of the proposed controllers and increase the applicability of the system.

**Author Contributions:** K.K.A. was the supervisor providing funding and administrating the project, and he reviewed and edited the manuscript. T.C.D. carried out the investigation, methodology, analysis, and validation, made the MATLAB and AMESim software, and wrote the original manuscript. D.T.T. carried out the simulations and checked the structure of the paper. Q.T.D. checked the manuscript and supported the model for research. All authors contributed to this article and accepted the final report. All authors have read and agreed to the published version of the manuscript.

**Funding:** This research was supported by Basic Science Program through the National Research Foundation of Korea (NRF) funded by the Ministry of Science and ICT, South Korea (NRF-2020R1A2B5B03001480).

**Conflicts of Interest:** The authors declare no conflict of interest.

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
