# Peer review of "Tracking Control for an Electro-Hydraulic Rotary Actuator Using Fractional Order Fuzzy PID Controller"

_electronics, doi:10.3390/electronics9060926_

Round 1

Reviewer 1 Report

This paper presents a strategy of fractional order fuzzy PID controller for trajectory-tracking control of an electro-hydraulic rotary actuator (EHRA) under variant working requirements. The proposed controller is based on a combination of a fractional order PID (FOPID)
16 controller and a fuzzy logic system.

In general, my suggestion is that a rigorous proof should be provided. Some minor comments are given as follows:

1. It is not easy to understand that Fig.3 is presented by a picture with a black background.

2. This reviewer feels that the proposed dynamics (8) can be stable by using traditional PID control laws or other intelligent control method, i.e, neural network based control laws. Why the authors use FOFPID controllers? The motivation should be emphasized.

3. The language of this paper needs further polishing, such as, "The results prove that..." should be "The results show/illustrate that..." in Line 78"; The sentence in Line 17 and 18 is not easy to understand.

4. The authors should explanation why the the chattering phenomenon exists?

5. From system (7) to system (8), the d_{2}(t) in system (8) is not defined or illustrated.

Author Response

Please see an attached file.

Reviewer 2 Report

In my opinion, the subject is interesting, however some issues should be added, explained or corrected:

A. Model issues:
1. can you justify a perfectly even division of the space of considerations into fuzzy sets (Fig.5) and the use of triangular functions?
2. can you explain inconsistency in fuzzy rule database of Ki:
- Ki: when e(t) is M, then increase in de(t) (S->M->B->VB) causes in general decrease in output (VB->VB->VB->B). However, when e(t) is B, then increase in de(t) (S->M->B->VB) causes in general INCREASE in output (S->S->M->M).
3. you use absolute values |e(t)| and |de(t)| in rule databases, which means that you assume symmetry of the system under and over the required working point. Do you think it is correct in hydraulic systems ?
4. can you explain one more issue: you calculate e(t) as the difference between required and the actual value of x1 (Eq.13). Hence, it can have either negative or positive value. However, you built your rule database on the basis of absolute value of e(t). So, for example in both cases: (1) when x1 is significantly LOWER than xref AND (2) when x1 is significantly HIGHER than xref, the |e(t)| is VB, and thus your output is VB or B. Do you think it will work properly ?
5. can you explain purpose of V5 and V6, non-controllable check valves ? What is their role in the system ? Under which condition there is a fluid flow through these valves ?
6. as far as I see, you didn't mention the defuzzification method, which is one of the most important elements of an FLC.
7. I also couldn't find any description of the used fuzzy operators (AND, OR) which you must have used, since your model has two inputs.

B. Editorial issues:
8. parameter designations should be consistent, if they are italic in the formulas, they should also be italic in the drawings (e.g. Fig. 1).
9. dots should be inserted in the points of crossings of hydraulic lines in Fig.1.,
10. How does Eq.4 come from Eq.3 (statement in line 108) ?
11. There are inconsistent denotations, e.g. you have U(n) in Eq.12, while there is u(t) in line 146. Should be corrected.
12. Fig.3 - I understand your intention to show increase in the controller range. However, solid fill makes it poor-readable. Maybe you should consider e.g. hatching instead of solid filling ? This would also enable you to show also a classic PID point (mi=1, lambda=1).
13. Table 4. (1) Effective bulk modulus has no unit (should be Pa). (2) What is "specific gravity" parameter of hydraulic oil ? Shouldn't it be "density" with unit "kg*dm^(-3)". (3) Unit of viscous friction coefficient. Is that correct ?
14. Fig.5 - the axes are not described.

Author Response

Please see an attached file.

Reviewer 3 Report

The article is at a good technical level. The topic of the article is current.

Author Response

Dear Reviewer 3,

Thank you for your recommendation.

Round 2

Reviewer 1 Report

All my concerns are addressed, and I have no further comments.

Reviewer 2 Report

Dear authors,
it seems that most of my suggestions have been included in the article.
In my opinion it may be printed in the journal.